# Graph Transformers Get the GIST: Graph Invariant Structural Trait for Refined Graph Encoding

## Abstract

Graph classification is a core machine learning task with diverse applications across scientific fields. Transformers have recently gained significant attention in this area, addressing key limitations of traditional Graph Neural Networks (GNNs), including oversmoothing and oversquashing, while leveraging the attention mechanism. However, a key challenge remains: effectively encoding graph structure information within the all-to-all attention mechanism, arguably the first step of all Graph Transformers. To address this, we propose a novel structural feature, termed Graph Invariant Structural Trait (GIST), designed to capture substructures within a graph through estimated pairwise node intersections. Furthermore, we extend GIST into a structural encoding method tailored for the attention mechanism in graph transformers. Our theoretical analysis and empirical observations demonstrate that GIST effectively captures structural information critical for graph classification. Extensive experiments further reveal that graph transformers incorporating GIST into their attention mechanism achieve superior performance compared to state-of-the-art baselines. These findings highlight the potential of GIST to enhance the structural encoding of Graph Transformers.

## 1. Introduction

Graph classification is a fundamental problem in machine learning with widespread applications in various domains, including chemistry, biology, and drug discovery (Dwivedi et al., 2022a;c; Irwin et al., 2012; Wu et al., 2017). The ability to classify graphs accurately enables advancements in predicting molecular properties, understanding complex biological interactions, and discovering novel therapeutic compounds. Traditional Graph Neural Networks (GNNs) (Kipf & Welling, 2017; Han et al., 2022) have been the cornerstone for such tasks, leveraging neighborhood aggregation to learn node and graph representations. However, GNNs often suffer from limitations such as oversmoothing (Keriven, 2022), oversquashing (Black et al., 2023), and restricted expressivity (Wang & Zhang, 2024) due to their reliance on local message-passing mechanisms.

Recently, Transformers (Vaswani et al., 2017) have emerged as a promising alternative for graph representation learning due to their global attention mechanism, which addresses many of the inherent limitations of GNNs. Transformers' ability to model complex interactions between entities makes them particularly attractive for graph classification (Ying et al., 2021). However, applying Transformers to graph data is not a seamless procedure, still posing unique challenges. Unlike sequential or image data, graph nodes typically lack inherent self-identity, making it difficult for Transformers to distinguish between entities purely based on their features. Without incorporating meaningful structural information, the attention mechanism in Transformers struggles to capture complex graph relationships effectively.

Existing approaches have attempted to improve Transformers with graph structural inductive bias by integrating positional or structural features, such as shortest path distances (Ying et al., 2021), Laplacian eigenvector-based encodings (Dwivedi et al., 2022a), and random walk-based features (Rampášek et al., 2022; Ma et al., 2023). While these methods provide some structural context, they either fail to capture comprehensive substructural information essential for distinguishing complex graph patterns (Rampášek et al., 2022) or focus predominantly on a limited set of substructures while neglecting higher-order structural relationships (Wollschlager et al., 2024). The challenge remains to identify a more expressive and comprehensive set of structural features, and devise efficient methods for encoding them within the Transformer's self-attention mechanism.

In this work, we introduce a novel structural feature called Graph Invariant Structural Trait (GIST), which captures the inherent substructures within a graph by estimating $k$-hop pairwise node intersections. Our approach is grounded in

[1]Anonymous Institution, Anonymous City, Anonymous Region, Anonymous Country. Correspondence to: Anonymous Author <anon.email@domain.com>.

Preliminary work. Under review by the International Conference on Machine Learning (ICML). Do not distribute.

the theoretical understanding that the cardinality of the intersection between two nodes' $k$-hop neighborhoods can serve as an effective permutation-invariant feature for substructure characterization, providing a robust foundation for graph classification. Incorporating GIST as a structural bias enhances the Transformer's capability to discern complex graph patterns, leading to improved classification performance. We further propose an efficient randomized algorithm to estimate GIST, ensuring scalability across large (number of) graphs. Through extensive experiments on various graph classification benchmarks, we demonstrate that integrating GIST into Graph Transformers achieves state-of-the-art performance and offers deeper insights into the structural properties of graph data.

Our key contributions are as follows:

- We introduce GIST, a method that encodes graph structure using pairwise $k$-hop substructure vector. These substructure vectors are efficiently computed by estimating the interaction cardinality between the $k$-hop neighborhoods of node pairs.
- We incorporate GIST into attention mechanisms of graph Transformers to enhance structural encoding. We provide both theoretical and empirical evidence demonstrating its effectiveness as a graph-invariant representation.
- We evaluate GIST-augmented graph Transformers on standard graph classification benchmarks, showing consistent performance improvements.

The introduction of GIST opens new avenues for enhancing the structural encoding capabilities of Transformers, paving the way for more effective and interpretable graph classification models.[1]

## 2. Motivation

Transformers, originally designed for sequential data, lack an inherent mechanism to capture the structural biases of graph data as highlighted in (Ying et al., 2021; Rampášek et al., 2022). Without a well-designed structural bias (structural encoding), they treat all nodes as equally related, failing to utilize the relational dependencies critical for graph tasks (Ying et al., 2021; Brody et al., 2022).

**Challenge 1. Capturing Graph Substructures in Structural Encoding.** The first key challenge in designing effective structural encodings for Graph Transformers is capturing the substructures within a graph, as these substructures often represent critical local patterns, or fragments that define the graph's overall characteristics (Ying et al., 2021; Ma et al., 2023; Wollschlager et al., 2024). While many early-stage structural encoding methods, such as shortest path distance (SPD) (Ying et al., 2021), provide a notion of

---

[1]The code will be made publicly available upon publication.

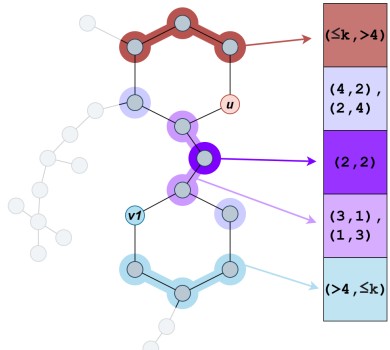

(a) $(u, v_1)$ from the same 6-ring substructure

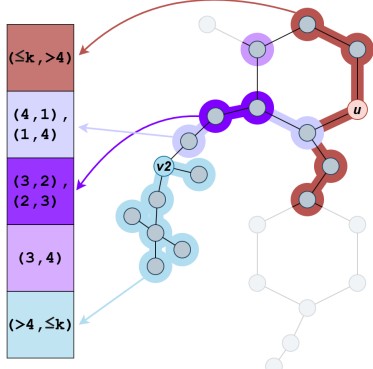

(b) $(u, v_2)$ from different substructures: a 6-ring and a 2-path

*Figure 1.* $k$-hop Substructure Vector Visualization (Def. 3.1) of ZINC molecule. The substructures of node pairs in the form of **intersection cardinality** of their common neighborhood at different distances from $u$ and $v$ are **"GIST"-ed into the Substructure Vector**. Specifically, each cell $(k_u, k_v)$ in the Substructure Vector denotes the number of nodes that are **exactly** $k_u$ hops from $u$ and $k_v$ hops from $v$. The variations in the Substructure Vector help the self-attention mechanism distinguish structural differences between node pairs, such as $(u, v_1)$ and $(u, v_2)$. For example, in Figure 1a, the pair $(u, v_1)$, which belongs to the **same** 6-ring substructure, has intersection cardinalities $\mathcal{I}_{(2,2)} = \mathcal{I}_{(4,2)} = \mathcal{I}_{(2,4)} = 1$. In contrast, the pair $(u, v_2)$, where $u$ and $v_2$ belong to **different** substructures (a 6-ring and a 2-path), has $\mathcal{I}_{(2,2)} = \mathcal{I}_{(4,2)} = \mathcal{I}_{(2,4)} = 0$.

proximity between nodes, they often struggle to effectively capture and represent substructures.

**Challenge 2. Aggregating Diverse Substructures Information.** As highlighted in (Wollschlager et al., 2024), it is equally important for structural encodings to enable the aggregation of information across diverse substructures, rather than restricting it to similar or localized patterns. Graphs, such as molecules, often exhibit a variety of substructures that interact in complex ways, and limiting information flow to nodes in different structures can hinder the model's ability to capture global dependencies and cross-pattern interactions. This is particularly important in domains like chemistry, biology, and social networks, where functional or structural properties often arise from specific subgraph

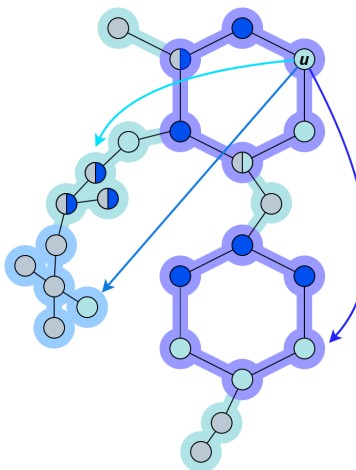

*Figure 2.* Node Clustering via Spectral Clustering Using Learned GIST Features in Graph Transformers on ZINC molecule graph. **Nodes within the same local substructures are clustered together**: 6-rings (purple), 2-path (cyan), and X-shape (light blue).

arrangements (i.e., rings and bonds in molecules) rather than the global graph structure alone (Yang et al., 2018; Yu & Gao, 2022). Many recent structural biases, such as shortest path distance (Ying et al., 2021) or those based on random walks (Rampášek et al., 2022; Ma et al., 2023), are effective at capturing simple substructures like cycles but tend to focus predominantly on these patterns, neglecting the interactions between different substructures (Wollschlager et al., 2024). For example, in Figure 2, it is more beneficial for $u$ to aggregate information from the 6-ring, X-shape, and 2-path substructures rather than solely focusing on another 6-ring that mirrors its own structural pattern. This highlights the need for a structural encoding that can help attention mechanisms effectively learn the substructures while enabling nodes to distinguish their own substructures from those of others, guiding attention based on the distinct structural relationships between nodes.

**Observation 1: Intersection Cardinality as a Discriminative Subgraph Feature.** Empirically, we observe that the intersection cardinality of common neighborhoods between two nodes $(u, v)$ can also serve as a powerful and discriminative feature encoding the $k-$hop subgraph structures. As illustrated in Figure 1, the intersections of common neighborhoods at different hop distances provide a structured way for $u$ to differentiate between the ring structure containing $v_1$ and the 2-path structure containing $v_2$, based on the differences in the in-between graph structures. Specifically, for $(u, v_1)$, which belongs to the same 6-ring substructure, the intersection cardinality values $\mathcal{I}_{(2,2)}$, $\mathcal{I}_{(4,2)}$, and $\mathcal{I}_{(2,4)}$ are all nonzero, indicating strong shared neighborhood connectivity. In contrast, $(u, v_2)$, which belongs to different substructures (a 6-ring and a 2-path), lacks these intersection values but instead exhibits nonzero intersection cardinality in positions such as $\mathcal{I}_{(3,2)}$ and $\mathcal{I}_{(2,3)}$, which are absent for

$(u, v_1)$. This contrast highlights how different substructure compositions lead to distinct intersection patterns, enabling the model to effectively distinguish between structurally similar and dissimilar node pairs, guiding the self-attention mechanism to weigh higher-order interactions accordingly.

**Observation 2: Intersection Cardinality Enhances Structural Awareness in Self-Attention Mechanisms.** Moreover, we empirically observe that incorporating an attention mechanism with intersection cardinality as an attention bias enables the attention mechanism to learn distinct substructures within the graph. In Figure 2, we train a Transformer architecture on the on ZINC dataset (Dwivedi et al., 2022a), introducing only the intersection cardinality (formally defined in Section 4 as GIST) as a bias in the attention scores. After training the model, we apply Spectral Clustering to group nodes based on the learned GIST features. The GIST features facilitate representation aggregation across structurally similar regions, allowing node $u$ to integrate information from another ring structure. This effect is evident as nodes from both rings are grouped into the same clusters, marked in dark blue and cyan. Furthermore, certain nodes positioned at the boundaries of these substructures act as "information exchange points", facilitating communication between distant regions of the graph. For example, the cyan-colored node within the "X" substructure is assigned to the same cluster as the ring nodes, effectively facilitating representation aggregation between two different substructures—an ability that current GNNs and Graph Transformers struggle with due to their inherent locality constraints. We note that this is **not a cherry-picked example**; rather, this phenomenon **consistently occurs across multiple samples** in the ZINC dataset after the Transformer is trained.

## 3. GIST: Graph Invariant Structural Trait

In this section, we formally introduce the graph invariant structural trait (GIST). We start by introducing how to encode the $k$-hop substructure of a node pair $(u, v)$ based on the $k$-hop common neighborhood between them. Next, we introduce how to use encoded $k$-hop substructures in a graph to form GIST. Finally, we introduce how to efficiently compute GIST with randomized hashing algorithms.

**Notation:** We denote an undirected graph $\mathcal{G} = (\mathcal{V}, \mathcal{E})$, which contains a set $\mathcal{V}$ of *n* nodes (vertices) and a set $\mathcal{E}$ of *m* edges (links). Each node $v \in \mathcal{V}$ has $d_n$ associated node features $x_v \in \mathbb{R}^{d_n}$, while each edge $e_{u,v} \in \mathcal{E}$ connecting node pair $(u, v)$ has $d_e$ associated edge features $y_{u,v} \in \mathbb{R}^{d_e}$ ($y_{u,v} = \mathbf{0}^{d_e}$ if there is no edge between $u$ and $v$). For every node $v \in \mathcal{V}$, we denote its $k$-hop neighborhoods as $\mathcal{N}_k(v)$. $\mathcal{N}_k(v)$ consists of all vertices that can be reached from $v$ with less or equal to $k$ edges. Subsequently, we define the $k$-hop common neighborhood of a node pair $(u, v)$ as

$\mathcal{C}_{k_u,k_v}(u,v) = \mathcal{N}_{k_u}(u) \cap \mathcal{N}_{k_v}(v)$, which is a set of nodes in the graph that can be reached within $k_u$ from $u$ and $k_v$ edges from $v$, respectively.

### 3.1. Encoding $k$-hop Substructure of a Node Pair

We encode the $k$-hop substructure of a node pair $(u,v)$ in a vector. This vector is computed based on the $k$-hop common neighborhood $\mathcal{C}_{k_u,k_v}(u,v)$.

**Definition 3.1** ($k$-hop substructure vector). Given a pair of node $(u,v) \in \mathcal{G}$, we propose capturing the $k-$hop graph structure between $u$ and $v$ with two types of features computed by $k$-hop common neighborhood $\mathcal{C}_{k_u,k_v}(u,v)$ as follows:

- $\mathcal{I}_{k_u,k_v}(u,v)$ as the cardinality of common neighborhoods that are exactly $k_u$ hops from node $u$ and $k_v$ hops from node $v$, computed as:

$$\mathcal{I}_{k_u,k_v}(u,v) = |\mathcal{C}_{k_u,k_v}(u,v)| - \sum_{\substack{x \leq k_u \,,\, y \leq k_v \\ (x,y) \neq (k_u,k_v)}} \mathcal{I}_{x,y}(u,v),$$

where $\mathcal{I}_{1,1}(u,v) = |\mathcal{C}_{1,1}(u,v)|$ for $u$ and $v$.

- $\mathcal{T}_{k_u}(u,v)$: the cardinality of nodes that are exactly $k_u$ hop from vertex $u$ and greater than $k$ hop from $v$ (and vice-versa for $\mathcal{T}_{k_v}(v,u)$), computed as:

$$\mathcal{T}_{k_u,k}(u,v) = |\mathcal{N}_{k_u}(u)| - \mathcal{T}_{k_u-1,k}(u) - \sum_{i=1}^{k_u}\sum_{j=1}^{k} \mathcal{I}_{i,j}(u,v)$$

For any node pair $(u,v)$, there would be $k^2$ numbers of $\mathcal{I}_{k_u,k_v}(u,v)$, $k$ numbers of $\mathcal{T}_{k_u,k}(u,v)$, and $k$ numbers of $\mathcal{T}_{k_v,k}(v,u)$. Finally, we encode the $k-$hop graph substructure surrounding node pair $(u,v)$ as a $k-$hop substructure vector $S_k(u,v)$. $S_k(u,v)$ starts with $\mathcal{I}_{k_u,k_v}(u,v)$ for every pair of $k_u, k_v \leq k$. Next, we fill the rest of the dimension in $S_k(u,v)$ with $\mathcal{T}_{k_u,k}(u,v)$ for each $k_u \leq k$ hop and $\mathcal{T}_{k_v,k}(v,u)$ for each $k_v \leq k$ hop.

As we see from Definition 3.1, computing the $k-$hop substructure vector requires first compute the cardinality of the $k$-hop common neighborhood $\mathcal{C}_{k_u,k_v}(u,v)$.

### 3.2. GIST: Graph Invariant Structural Trait

We define GIST as a three-dimensional matrix defined on the $k$-hop common neighborhood $\mathcal{C}_{k_u,k_v}(u,v)$ (see Definition 3.1) between every pair of node $(u,v)$ in graph $\mathcal{G}$.

**Definition 3.2** (Graph Invariant Structural Trait (GIST)). Let $\mathcal{G} = (\mathcal{V}, \mathcal{E})$ denote a graph with $n$ nodes ($|\mathcal{V}| = n$). We define the $k$-hop graph invariant structural trait (GIST) as a matrix $S(\mathcal{G}) \in \mathbb{R}^{n \times n \times (k^2+2k)}$, where each entry $S_{i,j}(\mathcal{G}) \in \mathbb{R}^{k^2+2k}$ is the $k$-hop substructure between node $v_i, v_j$ (see Definition 3.1). We also use $S(\mathcal{G})_{u,v}$ to represent the GIST value between node $u, v \in \mathcal{G}$.

GIST provides a compact representation of a graph's structural properties, encoding its topology and connectivity patterns by capturing higher-order relational dependencies among nodes and substructures. This encoding enables the differentiation of substructures, offering a detailed understanding of complex higher-order relationships, as illustrated in Figure 2 and Section 2. We would like to note one component of this representation: the diagonal entry $S_{i,i}(\mathcal{G})$, which essentially encodes the $k$-hop neighborhood surrounding a node $v_i \in \mathcal{V}$. This local structure provides a positional reference that differentiates nodes based on their placement within the global graph topology, enabling the model to capture long-range dependencies beyond direct connectivity. Mathematically, GIST represents pairwise node interactions as a matrix, where each interaction is encoded as a vector of dimension $(k^2+2k)$. This formulation preserves both local and global structural information, making GIST a comprehensive descriptor of graph architecture suitable for various analytical and learning-based applications.

### 3.3. Efficiently Compute GIST with Randomized Hashing

In this section, we show how to efficiently compute GIST by reducing the time complexity from $\mathcal{O}(k^2n^4)$ to $\mathcal{O}(k^2n^2)$. It is obvious that computing GIST $S(\mathcal{G})$ requires $\mathcal{O}(k^2n^4)$ time complexity. We note that for a node pair $(u,v)$, the exact computation of their $k$-hop common neighborhood $\mathcal{C}_{k_u,k_v}(u,v)$ incurs a cost of $\mathcal{O}(n^2)$, while calculating $S_{u,v}(\mathcal{G})$ requires $\mathcal{O}(k^2n^2)$. Consequently, computing $S_{u,v}(\mathcal{G})$ for all node pairs in a graph $\mathcal{G}$ results in an overall complexity of $\mathcal{O}(k^2n^4)$. Exact intersection calculations are computationally expensive, making them impractical for large graphs. Following (Chamberlain et al., 2022; Le et al., 2024), we propose to efficiently and unbiasedly estimate the cardinality of $k$-hop common neighborhood $\mathcal{C}_{k_u,k_v}(u,v)$ by decomposing it as:

$$|\mathcal{C}_{k_u,k_v}(u,v)| = \mathcal{J}_{k_u,k_v}(u,v) \cdot \mathcal{U}_{k_u,k_v}(u,v) \quad (1)$$

Here, $\mathcal{J}_{k_u,k_v}(u,v)$ represents the Jaccard similarity between $k_u$-hop neighborhoods $\mathcal{N}_{k_u}(u)$ and $k_v$-hop neighborhoods $\mathcal{N}_{k_v}(v)$. $\mathcal{U}_{k_u,k_v}(u,v)$ denotes the cardinality of the union $\mathcal{N}_{k_u}(u) \cup \mathcal{N}_{k_v}(v)$. Next, we can estimate $\mathcal{J}_{k_u,k_v}(u,v)$ with the constant-time collisions of the MinHash signatures of $\mathcal{N}_{k_u}(u)$ and $\mathcal{N}_{k_v}(v)$ as shown in Algorithm 1. We note that MinHash provides an unbiased estimator to the $\mathcal{J}_{k_u,k_v}(u,v)$ since the collision probability between the MinHash signatures of $\mathcal{N}_{k_u}(u)$ and $\mathcal{N}_{k_v}$ are equal to $\mathcal{J}_{k_u,k_v}(u,v)$ We can also estimate $\mathcal{U}_{k_u,k_v}(u,v)$ with the mergeable HyperLogLog sketch as Algorithm 1. We note that HyperLogLog also provides an unbiased estimator to $\mathcal{U}_{k_u,k_v}(u,v)$.

Finally, we multiply the estimated $\tilde{\mathcal{J}}_{k_u,k_v}(u,v)$ and

**Algorithm 1** Algorithm for computing intersection cardinality $|\mathcal{C}_{k_u,k_v}(u,v)|$

**Input:** Graph $\mathcal{G} = (\mathcal{V}, \mathcal{E})$, max hops $k$, hops $k_u, k_v$, $m$ MinHash functions $H = \{h_1, \ldots, h_m\}$, HyperLogLog parameter $p$ and regularizer constant $\alpha_p$
**Output:** Intersection cardinality $|\mathcal{C}_{k_u,k_v}(u,v)|$
{Step 1. Pre-compute MinHash signatures}
**for** $v \in \mathcal{V}, h_j \in H$ **do**
    $M_v[j, 0] \leftarrow h_j(v)$ {Initialize MinHash signatures}
**end for**
**for** $i = 1$ to $k$ **do**
    **for** $v \in \mathcal{V}, h_j \in H$ **do**
        $M_v[j, i] \leftarrow \min_{u \in \mathcal{N}(v)} \big(M_u[j, i-1], M_v[j, i-1]\big)$
    **end for**
**end for**
{Step 2. Pre-compute HyperLogLog sketches}
$m \leftarrow 2^p$
**for** $v \in \mathcal{V}$ **do**
    Compute $k$-hop HyperLogLog sketch $H_v \in \mathbb{R}^{m \times k}$
**end for**
{Step 3. Compute intersection cardinality}
**for** $(u, v) \in \mathcal{V} \times \mathcal{V}$ **do**
    $\tilde{\mathcal{J}}_{k_u,k_v}(u,v) \leftarrow$ JACCARD-EST$(k_u, k_v, m, M_u, M_v)$
    $\tilde{\mathcal{U}}_{k_u,k_v}(u,v) \leftarrow$ HLL-EST$(k_u, k_v, H_u, H_v)$
    $|\mathcal{C}_{k_u,k_v}(u,v)| \leftarrow \tilde{\mathcal{J}}_{k_u,k_v}(u,v) \cdot \tilde{\mathcal{U}}_{k_u,k_v}(u,v)$
**end for**
**return** $|\mathcal{C}_{k_u,k_v}(u,v)|$

**Function:** JACCARD-EST$(k_u, k_v, m, M_u, M_v)$
**Input:** hops $k_u, k_v$, number of MINHASH functions $m$, and $k-$hop MinHash values $M_u, M_v$
**Output:** Jaccard similarity $\tilde{\mathcal{J}}_{k_u,k_v}(u,v)$
$\tilde{\mathcal{J}}_{k_u,k_v}(u,v) \leftarrow 0$
**for** $j = 1$ to $m$ **do**
    **if** $M_u(j, k_u) = M_v(j, k_v)$ **then**
        $\tilde{\mathcal{J}}_{k_u,k_v}(u,v) \leftarrow \tilde{\mathcal{J}}_{k_u,k_v}(u,v) + 1$
    **end if**
**end for**
$\tilde{\mathcal{J}}_{k_u,k_v}(u,v) \leftarrow \tilde{\mathcal{J}}_{k_u,k_v}(u,v)/m$
**return** $\tilde{\mathcal{J}}_{k_u,k_v}(u,v)$
**EndFunction**

**Function:** HLL-EST$(k_u, k_v, H_u, H_v)$
**Input:** hops $k_u, k_v$, HyperLogLog sketches $H_u, H_v$
**Output:** Union cardinality $\tilde{\mathcal{U}}_{k_u,k_v}(u,v)$
$H_{k_u,k_v} \leftarrow \mathbf{0}^m$
**for** $j = 1$ to $m$ **do**
    $H_{k_u,k_v}[j] \leftarrow \max\big(H_u[j, k_u], H_v[j, k_v]\big)$
**end for**
$\tilde{\mathcal{U}}_{k_u,k_v}(u,v) \leftarrow \alpha_p m^2 (\sum_{i=0}^{m} 2^{-H_{k_u,k_v}[i]})^{-1}$
**return** $\tilde{\mathcal{U}}_{k_u,k_v}(u,v)$
**EndFunction**

$\tilde{\mathcal{U}}_{k_u,k_v}(u,v)$ together and form an unbiased estimator to $|\mathcal{C}_{k_u,k_v}(u,v)|$. This unbiased estimation can serve as an efficient alternative to exact computation for $|\mathcal{C}_{k_u,k_v}(u,v)|$. With MinHash and HyperLogLog, we reduce the computation time for $S_{u,v}(\mathcal{G})$ from $\mathcal{O}(k^2 n^2)$ to $\mathcal{O}(k^2)$, leading to $\mathcal{O}(k^2 n^2)$ time for compute GIST.

## 4. Graph Transformers Get the GIST

We see GIST can be naturally integrated into graph tansformers for graph structural encoding in the self-attention mechanism. As a result, we introduce the GIST attention for graph transformers.

**Definition 4.1** (GIST attention). Let $\mathcal{G} = (\mathcal{V}, \mathcal{E})$ denote a graph with $n$ nodes ($|\mathcal{V}| = n$). Let $x_u \in \mathbb{R}^{d_n}$ denote the representation of node $u \in \mathcal{V}$. Let $y_{u,v} \in \mathbb{R}^{d_e}$ denote the representation of edge between nodes $u, v \in \mathcal{V}$. Let $w_v \in \mathbb{R}^{d_n \times d_n}$ and $w_e \in \mathbb{R}^{d_n \times d}$ denote the model weight. Let $S(\mathcal{G})$ denote the $k$-hop GIST computed from $\mathcal{G}$ (see Definition 3.2). We define the GIST attention as a transform $\psi : \mathbb{R}^{d_n} \to \mathbb{R}^{d_n}$ on every node feature $x_u$ as:

$$\psi(x_u) = \sum_{v \in \mathcal{V}} \mathcal{A}_{u,v} \cdot (w_v x_v + w_e \hat{\mathcal{A}}_{u,v}),$$

where $\hat{\mathcal{A}}_{u,v} \in \mathbb{R}^d$ and attention score $\mathcal{A}_{u,v} \in \mathbb{R}$ are:

$$e_{u,v} = \phi_y(y_{u,v}) + \phi_S(S_{u,v}(\mathcal{G}))$$
$$\mathcal{A}_{u,v} = \sigma\big(\langle w_Q x_u + w_K x_v + w_b, e_{u,v}\rangle\big).$$
$$\hat{\mathcal{A}}_{u,v} = (w_Q x_u + w_K x_v + w_b) \odot e_{u,v}.$$

Here $\phi_y : \mathbb{R}^{d_e} \to \mathbb{R}^d$ and $\phi_S : \mathbb{R}^{k^2+2k} \to \mathbb{R}^d$ are MLP networks that align the representation of edge and GIST (see Definition 3.2) into same $d$-dimensional vector for addition. $w_Q, w_K \in \mathbb{R}^{d \times d_n}$ and $w_b \in \mathbb{R}^d$ are model weights and bias, respectively.

GIST attention can be viewed as a graph invariant with the following statement.

**Theorem 4.2** (Informal version of Theorem A.1). *Let $\mathcal{G} = (\mathcal{V}, \mathcal{E})$ denote a graph with $n$ nodes ($|\mathcal{V}| = n$). Let $S(\mathcal{G}) \in$ denote the $k$-hop GIST (see Definition 3.2) computed on $\mathcal{G}$. We show that the GIST attention (see Definition 4.1) $\psi(x_u)$ for every node $u \in \mathcal{V}$ is invariant under graph isomorphism.*

We provide the formal version of this theorem and proof in Appendix A. In other words, the permutation of node orders in the graph does not break the substructure in the graph due to graph isomorphism. As a result, it does not affect the value of GIST.

We use GIST attention as the building blocks and form a graph transformer with multiple GIST attention blocks. We view GIST attention as a way of modelling node interactions with the awareness of the graph structure.

# 5. Experiment

In this section, we aim to rigorously evaluate the effectiveness of GIST by addressing the following key research questions and providing corresponding insights:

- **RQ 1**: How well does GIST facilitate the learning and differentiation of substructures in graph classification tasks?
- **RQ 2**: To what extent does GIST enable long-range dependencies in Graph Transformers?
- **RQ 3**: How sensitive is GIST to the maximum hop distance for computing intersection cardinality?

## 5.1. Settings

We evaluate the proposed method on three benchmark suites comprising a total of 12 datasets, spanning small-scale to large-scale settings: the Long-Range Graph Benchmark (LRGB) (Dwivedi et al., 2022c), MoleculeNet (Wu et al., 2017), ZINC (Dwivedi et al., 2022a), and ZINC-full (Irwin et al., 2012). These datasets are specifically curated to emphasize challenges in structural encoding and long-range dependency modeling, with diverse applications in domains such as chemistry and biology.

**Baselines.** We benchmark the performance of our method against recent state-of-the-art baselines across multiple categories, including Graph Transformers, Graph Neural Networks (GNNs), hybrid models combining Transformers and GNNs, as well as pretrained graph models: GraphGPS (Rampášek et al., 2022), GRIT (Ma et al., 2023), Subgraphormer (Bar-Shalom et al., 2024), Frag-Net (Wollschlager et al., 2024), GatedGCN (Dwivedi et al., 2022c), SAN (Kreuzer et al., 2021), Graphormer (Ying et al., 2021), Graphormer-GD (Zhang et al., 2023b), GCN (Kipf & Welling, 2017), GIN (Xu et al., 2018), NGNN (Zhang & Li, 2021), DS-GNN (Bevilacqua et al., 2022), DSS-GNN (Bevilacqua et al., 2022), GNN-AK (Zhao et al., 2022), GNN-AK+ (Zhao et al., 2022), SUN (Frasca et al., 2022), OSAN (Qian et al., 2022), DS-GNN (Bevilacqua et al., 2023), GNN-SSWL (Zhang et al., 2023a), GNN-SSWL+ (Zhang et al., 2023a), GraphMVP (Liu et al., 2022), MGSSL (Zhang et al., 2021), and GraphFP (Luong & Singh, 2023).

**Experimental Settings.** For each dataset, we train our proposed method on the training set and select the epoch with the best validation performance. We then report the test results corresponding to this selected epoch. The performance of our method is presented as the mean ± standard deviation over 5 runs with different random seeds. The performance metrics for each baseline are obtained either directly from their original publications or reproduced by us using the best hyperparameters reported in their studies.

**Hyperparameters.** Particularly for our method, we perform a grid search to find the optimal hyperparameter combination for each dataset whenever feasible. The intersection

features are within [1,2,3,4,5,6]-hops of each node, the batch size is chosen among [32, 64, 128, 256], the number of layers is chosen among [2, 4, 6, 8], the number of heads is chosen among [2, 4, 8, 16, 32], the number of hidden dimensions is chosen among [16, 32, 64, 128], and learning rate is chosen among [0.0001, 0.0003, 0.0005, 0.002]. The chosen optimizer is AdamW. Our model is trained at 200 epochs for all datasets, except for MUV and HIV, where it is trained for 100 epochs. All model training and evaluations were conducted on NVIDIA A100 GPUs with 80G memory.

**Dataset Statistics.** We provide the statistics of 12 datasets used in our experiments to evaluate the performance of our proposed GIST in Table 1.

*Table 1.* Datasets' Statistics

| Dataset | # Graphs | Avg. # nodes | Avg. # edges | Prediction task | Metric |
|---|---|---|---|---|---|
| BBBP | 2,050 | 23.9 | 51.6 | binary classification | ROC-AUC |
| Tox21 | 7,831 | 18.6 | 38.6 | 12-task classification | ROC-AUC |
| Toxcast | 8,597 | 18.7 | 38.4 | 617-task classification | ROC-AUC |
| Sider | 1,427 | 33.6 | 70.7 | 27-task classification | ROC-AUC |
| Clintox | 1,484 | 26.1 | 55.5 | 2-task classification | ROC-AUC |
| Bace | 1513 | 34.1 | 73.7 | binary classification | ROC-AUC |
| MUV | 93,087 | 24.2 | 52.6 | 17-task classification | ROC-AUC |
| HIV | 41,127 | 25.5 | 54.9 | binary classification | ROC-AUC |
| Peptides-func | 15,535 | 150.94 | 307.30 | 10-task classification | Avg. Precision |
| Peptides-struct | 15,535 | 150.94 | 307.30 | 11-task regression | Mean Abs. Error |
| Zinc Subset | 12,000 | 23.2 | 49.8 | regression | Mean Abs. Error |
| Zinc Full | 249,456 | 23.2 | 49.8 | regression | Mean Abs. Error |

## 5.2. Long-Range Graph Benchmark (LRGB)

We evaluate the ability of our proposed GIST to learn long-range dependencies using two graph classification datasets from LRGB (Dwivedi et al., 2022c): Peptides-func and Peptides-struct. These datasets provide a robust benchmark for assessing graph classification methods in handling long-range dependencies and addressing structural challenges such as over-squashing and over-smoothing of many GNNs. As shown in Table 2, GIST significantly enhances the capability of Transformers, achieving state-of-the-art performance on LRGB. This demonstrates that encoding structural information into Transformer-based architectures can mitigate the limitations of existing GNNs in capturing long-range interactions. Regarding **RQ2**, our results demonstrate that GIST effectively captures long-range dependencies by encoding structural relationships beyond local neighborhoods, leading to improved classification performance.

## 5.3. ZINC and ZINC-full

We further evaluate our proposed GIST on two molecular property prediction datasets: ZINC (Dwivedi et al., 2022a) and ZINC-full (Irwin et al., 2012). These datasets are widely used benchmarks for assessing the ability of graph-based models to learn molecular representations and predict chemical properties. ZINC, with its constrained molecular structures and well-defined tasks, serves as a standard benchmark

*Table 2.* Performance of GIST on Peptides datasets from LRGB: Top-3 Results Highlighted in **Red**, **Blue**, and **Orange**.

| Model | Peptides-struct MAE ↓ | Peptides-func AP ↑ |
|---|---|---|
| GCN (Kipf & Welling, 2017) | $0.3496 \pm 0.0013$ | $0.5930 \pm 0.0023$ |
| GIN (Xu et al., 2018) | $0.3547 \pm 0.0045$ | $0.5498 \pm 0.0079$ |
| Subgraphormer (Bar-Shalom et al., 2024) | $0.2494 \pm 0.0020$ | $0.6415 \pm 0.052$ |
| FragNet (Wollschlager et al., 2024) | **$0.2462 \pm 0.0021$** | **$0.6678 \pm 0.0050$** |
| GatedGCN+RWSE (Dwivedi et al., 2022c) | $0.3357 \pm 0.0006$ | $0.6069 \pm 0.0035$ |
| GRIT (Ma et al., 2023) | **$0.2460 \pm 0.0012$** | **$0.6988 \pm 0.0082$** |
| GraphGPS (Rampášek et al., 2022) | $0.2500 \pm 0.0012$ | $0.6535 \pm 0.0041$ |
| SAN+LapPE (Kreuzer et al., 2021) | $0.2683 \pm 0.0043$ | $0.6384 \pm 0.0121$ |
| SAN+RWSE (Kreuzer et al., 2021) | $0.2545 \pm 0.0012$ | $0.6439 \pm 0.0075$ |
| GNN-SSWL+ (Zhang et al., 2023a) | $0.2570 \pm 0.006$ | $0.5847 \pm 0.0050$ |
| GIST (ours) | **$0.2442 \pm 0.0011$** | **$0.6783 \pm 0.0087$** |

for evaluating a model's effectiveness in capturing molecular topology and learning chemically relevant features. In contrast, ZINC-full provides a large-scale and more diverse dataset, offering a more rigorous test of a model's generalization capability across a broader range of molecular structures and chemical compositions. As shown in Table 3, our approach significantly improves the ability of Transformers to learn molecular graph representations, achieving superior predictive performance. These results demonstrate that incorporating structural priors into Transformer architectures can enhance molecular property prediction, making GIST a promising approach for advancing deep learning methods in computational chemistry and drug discovery.

*Table 3.* Performance of GIST on ZINC and ZINC-full: Top-3 Results Highlighted in **Red**, **Blue**, and **Orange**.

| Model | ZINC MAE ↓ | ZINC-full MAE ↓ |
|---|---|---|
| GCN (Kipf & Welling, 2017) | $0.367 \pm 0.011$ | $0.113 \pm 0.002$ |
| GIN (Xu et al., 2018) | $0.526 \pm 0.051$ | $0.088 \pm 0.002$ |
| NGNN (Zhang & Li, 2021) | $0.111 \pm 0.003$ | $0.029 \pm 0.001$ |
| DS-GNN (Bevilacqua et al., 2022) | $0.116 \pm 0.009$ | - |
| DSS-GNN (Bevilacqua et al., 2022) | $0.102 \pm 0.003$ | $0.029 \pm 0.003$ |
| GNN-AK (Zhao et al., 2022) | $0.105 \pm 0.010$ | - |
| GNN-AK+ (Zhao et al., 2022) | $0.091 \pm 0.002$ | - |
| SUN (Frasca et al., 2022) | $0.083 \pm 0.003$ | $0.024 \pm 0.003$ |
| OSAN (Qian et al., 2022) | $0.154 \pm 0.008$ | - |
| DS-GNN (Bevilacqua et al., 2023) | $0.087 \pm 0.003$ | - |
| GNN-SSWL (Zhang et al., 2023a) | $0.082 \pm 0.003$ | $0.026 \pm 0.001$ |
| GNN-SSWL+ (Zhang et al., 2023a) | $0.070 \pm 0.005$ | **$0.022 \pm 0.001$** |
| Subgraphormer (Bar-Shalom et al., 2024) | **$0.063 \pm 0.001$** | **$0.023 \pm 0.001$** |
| FragNet (Wollschlager et al., 2024) | $0.078 \pm 0.005$ | $0.024$ |
| GatedGCN-LSPE (Dwivedi et al., 2022c) | $0.090 \pm 0.001$ | - |
| GRIT (Ma et al., 2023) | **$0.059 \pm 0.002$** | **$0.023 \pm 0.001$** |
| GraphGPS (Rampášek et al., 2022) | $0.070 \pm 0.004$ | - |
| SAN (Kreuzer et al., 2021) | $0.139 \pm 0.006$ | - |
| Graphormer (Kreuzer et al., 2021) | $0.122 \pm 0.006$ | $0.052 \pm 0.005$ |
| Graphormer-GD (Kreuzer et al., 2021) | $0.081 \pm 0.009$ | $0.025 \pm 0.004$ |
| GIST (ours) | **$0.055 \pm 0.002$** | **$0.019 \pm 0.002$** |

### 5.4. MoleculeNet Benchmark

To further evaluate the effectiveness of our proposed GIST in molecular representation learning, we extend our exper-

iments to the MoleculeNet benchmark (Wu et al., 2017). MoleculeNet encompasses a diverse collection of graph-based molecular property prediction tasks, specifically designed to assess a model's ability to capture chemical interactions, molecular toxicity, and bioactivity. These tasks span a range of real-world applications, including drug discovery, environmental toxicity assessment, and material science, making MoleculeNet a comprehensive benchmark for evaluating graph-based learning approaches. As shown in Table 5, GIST consistently outperforms—or at least maintains competitive performance against—existing state-of-the-art pre-trained graph models and Graph Transformers across multiple tasks. These results highlight GIST's strong capability in molecular representation learning, demonstrating that structural information can be effectively integrated into Transformer-based architectures without the need for extensive pretraining, making it a promising approach for molecular property prediction in low-data regimes.

### 5.5. Ablation Study on different $k-$hop

Finally, to analyze the impact of different $k$-hop neighborhood sizes in our proposed GIST, we conduct an **ablation study** on the ZINC dataset. The value of $k$ influences how much local and long-range information is incorporated into the model. For **RQ3**, results from our ablation study on the ZINC dataset (Table 4) indicate that GIST is robust to variations in the maximum hop distance $k$. While performance improves as $k$ increases from 1 to 3, capturing richer structural dependencies, the fluctuations beyond $k = 3$ remain minimal, suggesting that GIST maintains stability across different neighborhood sizes. The slight decrease in performance at higher $k$ is marginal, indicating that GIST effectively balances local expressiveness and global aggregation without being overly sensitive to the choice of $k$.

*Table 4.* Ablation study on different $k$-hop neighborhood sizes in GIST on the ZINC dataset.

| $k$-**hop** | **1** | **2** | **3** | **4** | **5** |
|---|---|---|---|---|---|
| MAE ↓ | 0.100 | 0.058 | 0.054 | 0.065 | 0.063 |

For **RQ1**, our competitive results in Tables 5, 2, and 3 show that GIST effectively facilitates the learning and differentiation of substructures in graph classification tasks by encoding rich structural relationships through intersection cardinality. This enables Graph Transformers to capture fine-grained substructure information and complex substructure relationships, leading to improved performance.

## 6. Related Works

**Graph Substructures Modeling.** Modeling graph substructures is crucial for capturing fine-grained structural pat-

*Table 5.* Performance of GIST on MoleculeNet benchmark: Top-3 Results Highlighted in Red, Blue, and Orange.

| Model | BBBP | Tox21 | Toxcast | Sider | Clintox | Bace | MUV | HIV | Avg. AUC |
|---|---|---|---|---|---|---|---|---|---|
| AttrMasking (Hu et al., 2020a) | 64.3 ± 2.8 | 76.7 ± 0.4 | 64.2 ± 0.5 | 61.0 ± 0.7 | 71.8 ± 4.1 | 79.3 ± 1.6 | 74.7 ± 1.4 | 77.2 ± 1.1 | 71.2 |
| GRIT (Ma et al., 2023) | 69.9 ± 1.3 | 75.9 ± 0.6 | 65.6 ± 0.4 | 60.3 ± 1.2 | 85.9 ± 2.9 | 84.4 ± 1.2 | 77.1 ± 1.7 | 77.3 ± 1.5 | 74.8 |
| GraphGPS (Rampášek et al., 2022) | 56.2 ± 4.4 | 71.4 ± 0.7 | 60.6 ± 1.0 | 60.2 ± 1.1 | 79.2 ± 3.6 | 71.5 ± 6.0 | 65.2 ± 1.6 | 66.0 ± 9.4 | 66.3 |
| GraphLoG (Xu et al., 2021) | 67.8 ± 1.9 | 75.1 ± 1.0 | 62.4 ± 0.2 | 59.5 ± 1.5 | 65.3 ± 3.2 | 80.2 ± 3.5 | 73.6 ± 1.2 | 73.7 ± 0.9 | 69.7 |
| GraphCL (You et al., 2020) | 69.7 ± 0.7 | 73.9 ± 0.7 | 62.4 ± 0.6 | 60.5 ± 0.9 | 76.0 ± 2.7 | 75.4 ± 1.4 | 69.8 ± 2.7 | 78.5 ± 1.2 | 70.8 |
| G-Motif (Rong et al., 2020) | 66.9 ± 3.1 | 73.6 ± 0.7 | 62.3 ± 0.6 | 60.6 ± 1.5 | 77.7 ± 2.7 | 73.0 ± 3.3 | 73.0 ± 1.8 | 73.8 ± 1.2 | 70.2 |
| G-Contextual (Rong et al., 2020) | 69.2 ± 3.0 | 75.0 ± 0.6 | 62.8 ± 0.7 | 58.7 ± 1.0 | 60.6 ± 5.2 | 79.3 ± 1.1 | 72.1 ± 0.7 | 76.3 ± 1.5 | 69.3 |
| GPT-GNN (Hu et al., 2020b) | 64.5 ± 1.4 | 74.9 ± 0.3 | 62.5 ± 0.4 | 58.1 ± 0.3 | 58.3 ± 5.2 | 77.9 ± 3.2 | 75.9 ± 2.3 | 65.2 ± 2.1 | 67.2 |
| GraphFP (Luong & Singh, 2023) | 72.0 ± 1.7 | 74.0 ± 0.7 | 63.9 ± 0.9 | 63.6 ± 1.2 | 84.7 ± 5.8 | 80.5 ± 1.8 | 75.4 ± 1.9 | 78.0 ± 1.5 | 74.0 |
| MGSSL (Zhang et al., 2021) | 68.9 ± 2.5 | 74.9 ± 0.6 | 63.3 ± 0.5 | 57.7 ± 0.7 | 67.5 ± 5.5 | 82.1 ± 2.7 | 73.2 ± 1.9 | 75.7 ± 1.3 | 70.4 |
| GraphMVP (Liu et al., 2022) | 68.5 ± 0.2 | 74.5 ± 0.4 | 62.7 ± 0.1 | 62.3 ± 1.6 | 79.0 ± 2.5 | 76.8 ± 1.1 | 75.0 ± 1.4 | 74.8 ± 1.4 | 71.7 |
| GIST (ours) | 70.6 ± 1.8 | 77.2 ± 0.4 | 67.3 ± 0.9 | 61.3 ± 2.7 | 88.2 ± 2.2 | 86.0 ± 1.9 | 75.5 ± 3.2 | 77.0 ± 0.2 | 75.4 |

terns and improving representation learning in graph-based tasks. However, GNNs remain fundamentally constrained by their reliance on localized message passing, which limits their ability to capture long-range dependencies and effectively model complex substructure interactions, due to over-smoothing and over-squashing issues (Xu et al., 2018; Alon & Yahav, 2021). To address this, later works have introduced spectral features (Balcilar et al., 2021), motif-based methods (Rong et al., 2020; Zhang et al., 2021; Bar-Shalom et al., 2024; Wollschlager et al., 2024), and Weisfeiler-Lehman (WL) kernel-based approaches (Morris et al., 2019) to improve graph representation learning by explicitly capturing local and global structural patterns. While motif-based methods improve expressivity by incorporating recurring substructures, they often depend on predefined motifs, restricting their adaptability to unseen graph patterns. Similarly, WL kernel-based approaches enhance structural discrimination but struggle with distinguishing graphs that are structurally different yet WL-equivalent. Furthermore, spectral features capture global graph properties but introduce additional computational complexity, making them less practical for large-scale applications. These limitations underscore the need for alternative architectures that can more effectively integrate structural biases while maintaining both scalability and expressiveness in graph learning.

**Graph Transformers.** Transformers have demonstrated remarkable success in natural language processing and computer vision by leveraging self-attention to model long-range dependencies effectively (Vaswani et al., 2017). More recently, their adaptation to graph-structured data has led to the emergence of Graph Transformers, where self-attention replaces traditional message-passing mechanisms to enable more flexible and expressive learning (Zhang et al., 2020; Dwivedi & Bresson, 2021). However, a fundamental challenge in applying Transformers to graphs is the absence of a natural node ordering, making it difficult to encode structural information directly. To address this, positional encodings have been introduced to assign meaningful node representations within the graph topology. Among these, Lapla-

cian eigenvector-based encodings (LapPE) (Dwivedi et al., 2022a) and random walk positional encodings (RWPE) (Dwivedi et al., 2022b) inject global structural awareness, enhancing the model's ability to differentiate nodes with similar local neighborhoods. Beyond positional encodings, researchers have explored incorporating structural biases into self-attention to ensure that Graph Transformers respect the underlying graph topology. GPS (Rampášek et al., 2022) combines message passing with attention, allowing models to capture both local and global dependencies within the graph. More recently, GRIT (Ma et al., 2023) introduced a fully Transformer-based framework that eliminates explicit message passing while embedding structure-aware attention, achieving state-of-the-art performance across multiple graph learning benchmarks. These advancements reflect a growing shift toward pure Transformer architectures that effectively incorporate graph-specific inductive biases, paving the way for more scalable and expressive models in graph representation learning.

## 7. Conclusion

This paper introduces the Graph Invariant Structural Trait (GIST) to enhance Graph Transformers by improving their ability to encode graph structures. GIST estimates pairwise node intersections to capture substructures within a graph, integrating this information into the attention mechanism. This refinement enables Graph Transformers to better represent structural relationships that traditional all-to-all attention struggles to capture. Theoretical analysis and empirical results confirm that GIST effectively preserves essential structural information critical for graph classification. Extensive experiments across multiple datasets demonstrate that incorporating GIST into Graph Transformers consistently improves performance over state-of-the-art methods. These findings highlight the importance of structural encoding in enhancing Graph Transformers, contributing to more robust and interpretable graph-based learning models across scientific domains.

## Impact Statement

This paper presents work whose goal is to advance the field of Machine Learning. There are many potential societal consequences of our work, none of which we feel must be specifically highlighted here.

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

## A. Proofs

**Theorem A.1** (Formal version of Theorem 4.2). *Let $\mathcal{G} = (\mathcal{V}, \mathcal{E})$ denote a graph with $n$ nodes ($|\mathcal{V}| = n$). Let $S(\mathcal{G}) \in$ denote the $k$-hop GIST (see Definition 3.2) computed on $\mathcal{G}$. We show that the GIST attention $\psi(x_u)$ for every node $u \in \mathcal{V}$ (see Definition 4.1) is invariant under graph isomorphism.*

*Proof.* Let $f$ denote isomorphic transform on nodes $\mathcal{V}$ such that if $u$ and $v$ are adjacent in $\mathcal{G}$, $f(u)$ and $f(v)$ are also adjacent. Without loss of generally, we see that $\mathcal{C}_{k_u, k_v}(f(u), f(v)) = \mathcal{C}_{k_u, k_v}(u, v)$.

Following Definition 3.1, we show that $\mathcal{I}_{k_u, k_v}(f(u), f(v)) = \mathcal{I}_{k_u, k_v}(u, v)$, $\mathcal{T}_{k_u, k_v}(f(u), f(v)) = \mathcal{T}_{k_u, k_v}(u, v)$.

As a result, we show that $\mathcal{S}_{f(u), f(v)}(f(\mathcal{G})) = \mathcal{S}_{u, v}(f(\mathcal{G}))$.

Following Definition 4.1, since the order of node $v$ does not affect the computation of $\psi(x_u)$, we show that $\psi(x_{f(u)}) = \psi(x_u)$.

As a result, we show that the isomorphic transform $f$ does not change $\psi(x_u)$, making $\psi$ a graph invariant. $\qquad\square$

