# OpenReview forum: "Graph Transformers Get the GIST: Graph Invariant Structural Trait for Refined Graph Encoding"
_ICML.cc/2025/Conference — Submitted to ICML 2025_

### Official Review · Reviewer_1uvt · 2025-03-12

**Overall Recommendation:** 2

**Summary:**

This paper proposes a graph structural encoding method named Graph Invariant Structural Trait (GIST), aiming to improve Graph Transformers' ability to encode structural information. GIST captures structural features based on the intersection cardinality of pairwise nodes' k-hop neighborhoods. Empirical evaluations on multiple standard benchmarks demonstrate that integrating GIST into Graph Transformers enhances performance, surpassing several state-of-the-art models.

**Claims And Evidence:**

The claims made in the paper, particularly regarding the effectiveness of GIST in capturing complex substructures and long-range dependencies, are generally supported by experimental evidence. However, it seems that there is one paper (HDSE) that does the same thing [1]. I think this paper should at least be discussed in the paper.

[1] Enhancing Graph Transformers with Hierarchical Distance Structural Encoding.

**Essential References Not Discussed:**

[1] Enhancing Graph Transformers with Hierarchical Distance Structural Encoding.

**Experimental Designs Or Analyses:**

The experimental design of this paper is sound overall. The authors evaluate their proposed method across standard and relevant benchmarks.

**Methods And Evaluation Criteria:**

The chosen methods and benchmark datasets are appropriate for assessing graph classification capabilities.

**Other Comments Or Suggestions:**

See weaknesses.

**Other Strengths And Weaknesses:**

Strengths:

1. Clear exposition of the proposed model.
2. Extensive experiments and rigorous benchmarking.

Weaknesses (updated after rebuttal):

1. The current set of datasets is still limited; including additional LRGB and OGB datasets would enhance the comprehensiveness of the evaluation.

2. The experimental results would benefit from clearer visual examples.

3. The MoleculeNet benchmark setup omits critical details on reported baseline results like GRIT; without supplementary material or code, their validity cannot be verified.

**Questions For Authors:**

See weaknesses.

**Relation To Broader Scientific Literature:**

The paper’s key contributions closely relate to GRIT, Subgraphormer, and HDSE.

**Theoretical Claims:**

I examined the theoretical claims briefly.

---

> ### Author Rebuttal · Authors · 2025-04-01
>
> ### **[W1 - The theoretical analysis lacks clarity in terms of its practical implications for model improvements]**
>
> In Section 4, we present a theoretical analysis showing that GIST is invariant under graph isomorphism. This means GIST consistently captures the true structure of a graph, regardless of how its nodes are ordered. As a result, when used with a graph transformer, GIST prevents node ordering from influencing the model’s behavior.
>
> While the vanilla attention mechanism without positional encoding is also insensitive to node order, **incorporating traditional sequential position encodings reintroduces order sensitivity** as clearly stated in Graphormer by Ying et al., NeuRIPS 2021. This highlights the need for an attention bias like GIST that both encodes structural information and preserves invariance to node permutations.
>
>
> ### **[W2 - Datasets are not large-enough]**
>
> First, we would like to note our work **mainly focus on graph classification task**, and highlight a **potential discrepancy in the definition of a "large-scale graph classification dataset."** According to multiple recent works on graph transformers, the **ZINC-full dataset (250K graphs)** is widely regarded as a "large-scale" benchmark for graph classification and is one of the most commonly used datasets in this context. Here is supporting evidence:
>
> - **GRIT by Ma et al., ICML 2023**, a widely recognized SOTA graph transformer, states: *"We also conduct experiments on the larger datasets ZINC-full graphs (~250,000 graphs),"*
> - **FragNet by Wollschlager et al., ICML 2024**, a recent SOTA GNN-based method, explicitly refers to ZINC as a *"large-scale molecular benchmark."*
> - The **HDSE baseline by Luo et al., NeuRIPS 2024**, which was **suggested by the reviewer**, claims to apply *"HDSE to large-scale graphs."* However, their **largest graph-level task** is the Peptides dataset (~15K graphs), and they do not include ZINC-full in their experiments.
>
> We also surveyed recent graph transformer and graph-level works from **ICML 2024**, and to the best of our knowledge, **none of them included experiments on graph classification datasets larger than ZINC-full**. Given this context, we believe our choice of dataset is consistent with standard practices in the field.
>
> Nevertheless, we here provide additional experiment on PCQM4Mv2, **one of the largest-scale graph regression benchmark of 3.7M graphs**, from widely adopted OGB challenge in Table 1. Due to time constraints and the large size of **PCQM4Mv2**, GIST's training has not yet fully converged. Nevertheless, the current results already surpass many baselines, and we anticipate even stronger performance as training continues over the next few days. We also provide experiment results one two node classification datasets to showcase the applicability of our proposed method to different graph tasks.
>
> **Table 1**: Performance of GIST on Cluster, Pattern, and PCQM4Mv2 datasets.
> | Datasets | Cluster | Pattern | PCQM4Mv2 |
> |-|:-:|:-:|:-:|
> | GIST | **0.7906** | **0.8693** | **0.089** |
> | GPS | 0.7802 | 0.8668 | 0.094 |
> | SAN | 0.7669 | 0.8658 | - |
> | GatedGCN | 0.7384 | 0.8557 | - |
>
> ### **[W3 - Essential discussion of related work, HDSE]**
>
> We thank the reviewer for their meticulous review of our work and agree that HDSE follows a similar trajectory in capturing substructures within graphs. However, we would like to highlight that HDSE's structural bias, **Graph Hierarchy Distance**, differs from our proposed bias, **Graph Invariant Structural Trait (GIST)**. While both approaches enhance graph transformers, **GIST consistently outperforms HDSE across various datasets (Table 2)**. We attribute this to GIST’s ability to capture **higher-order structural relationships** through the **information exchange point**, as discussed in **Observation 2 of the Motivation section**. We will make sure to incorporate this discussion into the final version of our paper.
>
> **Table 2**: Performance comparison of GIST vs. HDSE
> | Methods | ZINC | Peptides-struct | Peptides-func |
> |-|:-:|:-:|:-:|
> | GIST | **0.055** | **0.2442** | 0.6783 |
> | HDSE | 0.059 | 0.2457 | **0.7156** |
>
> ### **[W4 - Visualization for experimental results]**
>
> Our tables and **color-coded rankings (top-1,2,3)** follow **standard practices in the field—including the HDSE baseline suggested by the reviewer**.
>
> Since the suggestion *"The experimental results could be supplemented with visual examples"* is broad, we are unsure what specific visualization is desired. If the reviewer means bar charts, we are happy to provide them—otherwise, please clarify, and we will accommodate accordingly.
>
> We hope our rebuttal clarifies concerns on dataset scale, related work, and results. We’ve added PCQM4Mv2 experiments to demonstrate scalability and are open to further experiments or visualizations as needed. Given these improvements, may we kindly ask the reviewer to reconsider our contributions and rating?

---

> > ### Comment · Reviewer_1uvt · 2025-04-04
> >
> > Thanks for your careful response. Some key questions still remain for me:
> >
> > **(1) Datasets are not large-enough**
> >
> > Thank you for the additional experiments. I did not mean to suggest using the OGB PCQM4Mv2 dataset, as it is indeed too large and likely infeasible given time constraints. Rather, since you have already included *Peptides-struct* and *Peptides-func*, I was wondering why other datasets from the LRGB benchmark—such as *PascalVOC-SP*, *COCO-SP*, and *PCQM-Contact*—were not considered. Some datasets from OGB could also be relevant. Including a broader range of datasets would strengthen the empirical evaluation and better demonstrate the generality of your method.
> >
> > **(2) Essential discussion of related work**
> >
> > The paper claims to compare against *state-of-the-art baselines* for graph classification. However, in the current version, GRIT (a 2023 method) is mentioned as such. There are several 2024 models that achieve strong performance on benchmark datasets and should at least be acknowledged when discussing the state-of-the-art. For example:
> >
> > - [1] reports 0.012 MAE on ZINC-Full,
> > - [2] reports 0.014 MAE on ZINC-Full,
> > - [3] reports 0.046 MAE on ZINC, and
> > - [4] reports 0.7311 AP on Peptides-func.
> >
> > I don’t mean to suggest that your method needs to outperform these baselines, but rather that referencing some recent strong methods would make the claim of state-of-the-art comparison more balanced and complete.
> >
> > [1] An end-to-end attention-based approach for learning on graphs, arXiv, Feb 2024
> >
> > [2] Topology-Informed Graph Transformer, arXiv, Feb 2024
> >
> > [3] Graph Attention with Random Rewiring, arXiv, Jul 2024
> >
> > [4] Spatio-Spectral Graph Neural Networks, NeurIPS 2024
> >
> > **(3) Visualization for experimental results**
> >
> > Apologies if my previous comment was unclear. Since GIST can be naturally integrated into graph transformers, I suggest providing visualizations that illustrate how attention patterns change after incorporating GIST. This could help readers better understand the mechanism and benefit of the integration. (In the reply, the provided visualization was difficult to interpret—please clarify how these visualizations were generated and explain the specific role and significance of GIST in them)
> >
> > **(4) Concerns Regarding MoleculeNet Benchmark Results**
> >
> > Upon reviewing Section 5.4 in light of the authors’ rebuttal regarding graph-level tasks, I noticed that the experimental setup for the MoleculeNet benchmark lacks sufficient detail. Specifically, the source of the reported baseline results—GRIT on datasets such as BBBP, Tox21, ToxCast, SIDER, ClinTox, BACE, MUV, and HIV—is unclear. The original GRIT paper does not report results on these datasets, and *since no supplementary material or code was provided*, it is difficult to verify how these results were obtained. (April 7 edit)
> >
> > Overall, I find this work promising, and with a more thorough analysis of GIST and a clearer positioning in relation to recent literature, I believe it can be significantly strengthened.

---

> > > ### Author Response · Authors · 2025-04-06
> > >
> > > ### **`1. "I was wondering why other datasets from the LRGB benchmark—such as...—were not considered."`Because they are not graph classification datasets, nor are they considered very large-scale.**
> > >
> > >
> > > We thank the reviewer for the clarification. We did not include coverage for the 3 LRGB datasets the reviewer mentioned because:
> > >
> > > - **They are not graph classification ones** — the domain focus of our work, which we have already featured all graph classification datasets within LRGB.
> > > - **Nor are they considered very large-scale** — see below.
> > >
> > > **T1: Dataset Stats**
> > > |Dataset|# of Graphs|
> > > |-|-|
> > > |PascalVOC-SP|11,355|
> > > |COCO-SP|123,286|
> > > |ZINC-full|249,456|
> > > |PCQM4Mv2|3,746,619|
> > >
> > > Thus, when we initially saw the reviewer comment:
> > >
> > > > W2 ***"The study lacks experiments on large-scale datasets."***
> > >
> > > **we naturally turned to one of the largest graph-level datasets, OGB PCQM4Mv2, to meet that concern**. Especially since we already included **ZINC-full**, a large-scale benchmark used in established works like GRIT and FragNet.
> > >
> > > ---
> > >
> > > That said, **we appreciate the opportunity to further demonstrate GIST's generalizability beyond graph classification**. Below, we present **results on PascalVOC-SP (as requested)**, along with two additional **node-level datasets** to cover a broader range of tasks.
> > >
> > > **Table 2: GIST on Node-Level Tasks**
> > > |Dataset|Cluster|Pattern|PascalVOC-SP|
> > > |-|-|-|-|
> > > |GIST|**0.7906**|**0.8693**|**0.3789**|
> > > |GPS|0.7802|0.8668|0.3748|
> > > |SAN|0.7669|0.8658|0.3230|
> > > |GatedGCN|0.7384|0.8557|0.2873|
> > >
> > > We hope these results further highlight the versatility and effectiveness of GIST.
> > >
> > > ---
> > >
> > > Moreover, since we already invested significant effort and compute into **PCQM4Mv2**, we may as well present our final results on this:
> > >
> > > **Table 3: GIST on PCQM4Mv2**
> > > |Dataset|PCQM4Mv2|
> > > |-|-|
> > > |GIST|**0.0844**|
> > > |GPS|0.0852|
> > > |GRIT|0.0859|
> > >
> > > We also refer the reviewer to our responses to **W7&8 from reviewer `nG8X`** for a broader discussion on generalization. In total, we have evaluated our method on **12 datasets in the main paper** and **4 additional datasets during the rebuttal** covering tasks spanning graph-level, node-level, long range, and large-scale scenarios. **We believe it is fair to argue results on such 16 datasets present a thorough and well-rounded evaluation of GIST and is well beyond the coverage standard done by most related work (and acknowledge by all other reviewers)**. We sincerely hope t reviewer will see our virtue in this regard, too.
> > >
> > > ---
> > >
> > >
> > >
> > >
> > > ### **`2. "The paper claims to compare against state-of-the-art baselines for graph classification. However, in the current version, GRIT (a 2023 method) is mentioned as such."` Sorry for the nitpick, but we never claimed (or at least not meant to claim) GRIT as the sole SOTA baseline. But we will certainly add more discussion about such works.**
> > >
> > > Around `L298 - L315`, our writing reads:
> > >
> > > > *We benchmark the performance of our method against recent state-of-the-art baselines across multiple categories, including Graph Transformers, Graph Neural Networks (GNNs), hybrid models combining ... as well as pretrained graph models: <works listed>*
> > >
> > > **Where we have then listed around multiple works as SOTA baselines.** Though we did mention GRIT as SOTA at a few times, we never meant it is the only SOTA baseline. On the broader scale, while we indeed did not feature [1-4], we argue we featured works with similar recency (e.g., Subgraphormer & FragNet), making up a fair representation of *SOTA baselines for graph classifications.* And we hope being an understanding reviewer as you are, you would see our perspective, especially given [1, 2] have no cited opensourced implementations and [3] is just accepted at ICLR.
> > >
> > > **That said, we again appreciate the opportunity to discuss more work as non-baseline but related work, and we find many suggestions from the reviewer particualrily good.** We plan to add the following discussion (with much more details) in the updated version:
> > >
> > > > SE2GNN [4] and TIGT [2] follow a similar trajectory as Subgraphormer by enhancing GNNs with substructure awareness. SE2GNN tackles the long-range aggregation problem using global spectral filters. The key distinction between these methods and GIST lies in the backbone architecture—GNNs vs. transformers. GRASS [3] extends GRIT by incorporating random rewiring, while ESA [4] proposes a new graph transformer architecture. Both GRASS and ESA are orthogonal to our method, where interesting combinations shall be explored.
> > >
> > > We hope the reviewer would find it helpful.
> > >
> > > ---
> > >
> > >
> > >
> > > ### **`3. "I suggest providing visualizations that illustrate how attention patterns change after incorporating GIST."` Sure!**
> > >
> > >
> > > We don't have much char left so pls allow us to be brief and direct: Thank you for clarifying, it is a great suggestion and we are impressed by similar vis done in HDSE. We follow a similar style and present our vis here: https://anonymous.4open.science/r/GIST_Visualization-B756/README.md.
> > >
> > > Thanks again!

---

### Official Review · Reviewer_e5VJ · 2025-03-13

**Overall Recommendation:** 3

**Summary:**

In this paper, authors propose a new Graph Invariant Structural Trait (GIST) for higher-order structural relationship modeling within graphs and utilize randomized hashing to accelerate the corresponding calculation. The usage of GIST in the graph transformer has proven to be effective through experiments on several datasets.

**Claims And Evidence:**

Yes. Most of the claims are reasonable and clear.

**Essential References Not Discussed:**

No, according to my knowledge, related works have been well cited.

**Experimental Designs Or Analyses:**

Yes. Authors conduct extensive experiments on different datasets and provide corresponding ablation study and analysis. However, since the calculation of GIST seems to bring considerable amount of computations, authors should also include the efficiency metrics in the performance comparison.

**Methods And Evaluation Criteria:**

Yes.

**Other Comments Or Suggestions:**

No.

**Other Strengths And Weaknesses:**

Strengths:
1. The design of GIST is reasonable, which can capture the inherent structures within the graph.
2. Authors provide detailed experiments to evaluate their design.

Weaknesses:
1. Whether GIST has practical value remains questionable since it seems to introduce extra calculation overhead. Authors can provide the execution time comparison to demonstrate its efficiency.
2. The writing of the paper can be improved; unclear descriptions should be avoided. E.g., on Page.3, “each node has d_n associated node features”, it’s ambiguous whether the number of features or the dimension of features is d_n. Besides, Fig.1 is very hard to understand without enough explanation.

**Questions For Authors:**

No.

**Relation To Broader Scientific Literature:**

Compared to previous methods, GIST explicitly models the higher-order structural information within the graph by estimating k-hop pairwise node intersections, which is helpful for accurately capturing the various substructures.

**Theoretical Claims:**

Authors should provide more detailed calculation process for the complexity claimed in Sec.3.3.

---

> ### Author Rebuttal · Authors · 2025-04-01
>
> ### **[W1 - Exra overhead cost of GIST computation]: Sure. Here are an analysis of GIST computation overhead cost and training efficiency of our proposed method.**
>
> We kindly direct them to our new analysis on the one-time pre-computation overhead of GIST features and the overall training efficiency of GIST in our response to W2 of reviewer `L3nR`. We would like to emphasize that **GIST computation incurs only a one-time cost at the beginning**, and this overhead is minimal. We also provide experimental results of GIST on **PCQM4Mv2 (~3.7M graphs), one of the largest graph-level challenges in the community**, to further reinforce its scalability and efficiency in our response to W2 of reviewer `1uvt`.
>
> ### **[W2 - Typo. Figure 1 could be difficult to interpret.]: May we kindly ask which part of Figure 1 is confusing the reviewer? In the mean time, we elaborate Figure 1 again here.**
> We thank the reviewer for their careful reading of our paper. We will clarify any ambiguous technical terms in the final version. Regarding the specific points raised:
>
> 1. The statement **"Each node has d_n associated node features"** is intended to mean **"Each node has d_n-dimensional associated node features,"** which we specify mathematically as **"$x_v \in \mathbb{R}^{d_n}$"**. We will make this and related definitions clearer in the final version.
> 2. Since we have provided a **13-line explanation** in the caption of **Figure 1** and further details in the **Motivation section (Observation 1)**, may we kindly ask the reviewer to specify which part is unclear so that we can provide a more precise clarification?
> 3. While reiterating the purpose of **Figure 1** poses a risk of redundancy, we are happy to elaborate again here: Using the same graph, subfigure **1a** illustrates **a part of the 4-hop GIST features** for two nodes *$(u, v_1)$* that belong to the **same substructure**. In contrast, subfigure **1b** depicts **a part of the 4-hop GIST features** for two nodes *$(u, v_2)$* that belong to **different substructures**. The GIST feature is a tensor where each cell **$(k_u, k_v)$** encodes the number of nodes in the neighborhood that are exactly **$k_u$** hops from node **$u$** and **$k_v$** hops from node **$v$**. This comparison highlights how GIST features enable the transformer to **distinguish nodes belonging to different substructures** within the same graph, guiding the attention mechanism to differentiate node pairs more effectively. Please refer to our *Observation 1 in Motivation section* for a more detailed discussion. We hope this clarifies the reviewer’s concerns regarding **Figure 1**.
>
> ### **[Sec 3.3 clarification]**
>
> We further elaborate on the **theoretical complexity** of **exact GIST feature computation**, which we addressed in **Section 3.3** by introducing an efficient **estimation algorithm** using MinHash and HyperLogLog. For empirical validation of the algorithm's efficiency, please refer to our response to W2 of reviewer `L3nR` for detailed results.
>
> A **naive approach** to compute the **exact** number of nodes in the **$(k_u, k_v)$-neighborhood intersection** for a node pair $(u,v)$, denoted as $C_{k_u,k_v}(u,v)$, follows the pseudocode below:
>
> ```pseudo
> counts = 0
> FOR x_u in N_ku(u) do
>     FOR x_v in N_kv(v) do
>         if x_u == x_v do
>             counts += 1
> RETURN counts
> ```
> ### **Time Complexity Analysis:**
> 1. The **worst case** (e.g., a fully connected graph) results in a worst-case complexity of **$O(n^2)$** for computing a single $C_{k_u,k_v}(u,v)$.
> 2. Since GIST requires computing $k^2$ such values per node pair $(u,v)$, the complexity increases to **$O(k^2 n^2)$** per node pair.
> 3. Given that a graph $G$ with $n$ nodes has **$n^2$ node pairs**, the total complexity becomes **$O(k^2 n^4)$** for exact GIST computation, which is impractical for a large number of graphs.
>
> To **overcome this infeasibility**, we propose a **low-complexity estimation algorithm** (**Algorithm 1** of our paper) to approximate **GIST features efficiently**. We highlight that the **naive** $O(n^2)$ **approach** to compute the **exact** number of nodes in the **$(k_u, k_v)$-neighborhood intersection** for a node pair $(u,v)$ can be **efficiently estimated in constant time** $O(1)$ using Algorithm 1. With that, we eliminate the **$O(n^2)$ bottleneck** from the overall complexity. As a result, the final time complexity for computing **GIST features across the entire graph** is reduced to **$O(k^2 n^2)$**, making our method significantly more scalable for a large number of graphs.
>
> We hope the additional results and discussion provide the reviewer with a clearer understanding of the mechanism behind our proposed method. Given these clarifications and improvements, we kindly ask the reviewer to consider whether our contributions warrant a higher rating.

---

> > ### Comment · Reviewer_e5VJ · 2025-04-02
> >
> > Thanks for the efforts. So, in Fig.1, $I(2, 2)$ means $I_{2, 2}(u, v)$ as introduced in Sec.3.1, authors should keep the consistency of the expression and avoid unnecessary omissions. Anyway, additional information has addressed most of my concerns. As for the efficiency problem, I am more concerned about the prediction time comparison with baselines, rather than the precomputing and training time alone. To sum up, I am willing to improve the rating from 2 to 3.

---

> > > ### Author Response · Authors · 2025-04-04
> > >
> > > ### **`W1 - GIST inference efficiency (prediction time):` Sure, here we provide comparison of inference time between GIST and baselines.**
> > >
> > > We thank the reviewer for acknowledging the merit of our method and raising the score from 2 to 3. We also appreciate the reviewer’s clarification on inference efficiency. **Table 1** below reports the inference time of **GIST and other baselines** across four datasets, where inference time is **a one-time structural encoding pre-processing time + model prediction time for a single batch of size 32 (zinc) or of size 16 (petides-struct, func)**. For any given graph—whether in training or testing—GIST features require only a one-time precomputation. Our results show that **GIST’s inference time is on par with other graph transformers**, demonstrating its efficiency in real-world applications.
> > >
> > > **Table 1**: Inference Time (in seconds)
> > > | Datasets | ZINC | ZINC-full | Peptides-struct | Peptides-func |
> > > |-|:-:|:-:|:-:|:-:|
> > > | GIST | 0.3 | 0.3 | 0.7 | 0.7 |
> > > | GRIT | 0.03 | 0.03 | 0.1 | 0.1 |
> > > | HDSE | 0.42 | 0.42 | 1.1 | 8.6 |
> > >
> > > We will ensure that all these thoughtful discussions and suggestions from the reviewer (e.g., notation consistency) are incorporated into the later version. While the reviewer has already improved the rating—which we sure appreciate—we shamelessly venture to ask for a further improvement if inference efficiency is the only remaining concern; as we believe it should be well-addressed with the inference results above.
> > >
> > > ---
> > >
> > > Last, we want to take this opportunity to highlight **someconcerns raised by multiple reviewers that have already been acknowledged by reviewer `e5VJ`**, such as:
> > > - **W1 on "one-time computation overhead" and "inference efficiency"**: Both `L3nR` and `e5VJ` raised similar concerns about GIST’s efficiency. We thank `e5VJ` for recognizing the effectiveness of our estimation algorithm in mitigating GIST's pre-computation overhead. We hope that our not supplied inference time results further clarify GIST’s **practical viability**, given its minimal additional overhead and efficient end-to-end performance.

---

### Official Review · Reviewer_nG8X · 2025-03-13

**Overall Recommendation:** 3

**Summary:**

Graph classification, a fundamental machine learning task with broad scientific applications, has been advanced by Transformers, which address oversmoothing/oversquashing limitations of traditional GNNs using attention mechanisms. However, effectively encoding graph structural information within Transformers' all-to-all attention remains challenging. To tackle this, the proposed Graph Invariant Structural Trait (GIST) captures critical substructure details via pairwise node intersections, enhancing structural encoding in graph Transformers and outperforming state-of-the-art methods in experiments.

**Claims And Evidence:**

Yes.

**Essential References Not Discussed:**

No.

**Experimental Designs Or Analyses:**

Yes.

**Methods And Evaluation Criteria:**

Yes.

**Other Comments Or Suggestions:**

See my comments above.

**Other Strengths And Weaknesses:**

Strengths:
1. developing expressive graph transformers is important
2. the paper is well written
3. experiments show the method outperforms the baselines

Weaknesses:
1. how to guarantee the method can ggregating diverse substructures information
2. how to theorectically verify the proposed method could be more expressive than other methods
3. better to use larger-scale datasets
4. how does the theorectial analysis contributes? Simple methods like transformers without position encodings could also produce graph-invariant representations. So do set-like methods.
5. how about hyperparameter sensitivity?
6. Could it be combined with pther structural encodings?
7. Could it be applied in GraphLLM or GFM?
8. Could it be applied to node or link-level tasks?

**Questions For Authors:**

See my comments above.

**Relation To Broader Scientific Literature:**

No.

**Theoretical Claims:**

Yes.

---

> ### Author Rebuttal · Authors · 2025-04-01
>
> We thank the reviewer for the detailed feedback. Given the 5k char limitation and 8 distinct questions raised by the reviewer, unfortunately many of our responses will be condensed and citing other replies. Should the reviewer be interested in an elaboration in any particular response, please let us know and we will be more than happy to accomodate.
>
> ### **[W1 - How to guarantee the method can aggregate diverse substructures information]**
>
> GIST encodes $k$-hop substructures between node pairs (Definition 3.2), enabling a broad representation of **substructural relationships** through **all-to-all comparisons**.
>
> When integrated with graph transformers, GIST’s all-to-all substructure information enhances the attention mechanism. Specifically, with GIST, each node’s embedding is updated by incorporating **structural interactions with every other node**, allowing for a more structure-aware aggregation of representations. This facilitates **effective propagation of structural patterns** across the graph.
>
> As shown in Figure 1, GIST helps differentiate substructures, guiding **diverse attention aggregation**—a capability further visualized in Figure 2 with learned GIST features on **ZINC**.
>
> ### **[W2 - How to theoretically verify the proposed method could be more expressive than other methods]**
>
> It is difficult to **rigorously prove** that one graph feature is universally more expressive than another. For instance, Proposition 3.2 in GRIT shows that GD-WL with RRWP is **at least as expressive as** GD-WL with SPD and highlights a **special case** where RRWP outperforms SPD. However, this does **not** imply that RRWP is **strictly superior** across all scenarios—only in rare, impractical graph structures.
>
> This underscores a broader issue: *theoretical arguments on graph feature expressiveness often lack generality and rigor*, even in existing works. Thus, we prioritize **empirical analysis** to assess practical effectiveness, a stance **acknowledged by reviewers `e5VJ` and `1uvt`**.
>
> ### **[W3 - Datasets are not large enough]**
> Please refer to **W2 in our response to reviewer `1uvt`** for a detailed discussion on large-scale datasets.
>
> ### **[W4 - Importance of structural-invariance in GIST]**
>
> We refer reviewer nG8X to our detailed discussion on the importance of structural-invariance in GIST in our response to W1 of reviewer `1uvt`.
>
>
> ### **[W5 - Hyperparameter sensitivity analysis]**
>
> We direct the reviewer to our new ablation studies in our response to W2 of reviewer `L3nR`.
>
> ### **[W6 - Combination with other structural encoding]**
>
> We have already explored the combination of GIST with RRWP and SPD previously, but neither improved performance. Here, we present one result on **GIST + RRWP**, where RRWP is concatenated with GIST as a structural bias:
>
> $[x \| y] \in \mathbb{R}^{k^2 + 2k + \text{steps}}$
>
> As shown in **Table 5** in our response to reviewer `L3nR`, **GIST alone outperforms the combination**. While both encode structural information, GIST has been shown empirically to be a stronger structural bias. Thus, incorporating RRWP may introduce noise into GIST’s learning process, leading to a slight drop in transformer performance.
>
>
> ### **[W7,W8 - Application to GraphLLM, GFM, node- and link-level]**
>
> We are not too sure whether the reviewer is expecting some Yes/No answers or is actually requesting us to do it. In short, **the  answers to these questions are generally a "Yes, GIST has the potential to extend to [x]"; but, respectfully, we believe it can be fairly argued that such applications are clearly out of scope.** This is evident by the fact that most established prior works on graph transformers — such as GRIT, HDSE, and Subgraphormer — do not explore such extensions like GLLM/GFM.
>
> The underlying reason is such an extension would very much deserve a paper of its own, potentially requiring extensive pre-training and pipeline efforts  and being mindful of typical GFM challenges like dim mismatches and hyperspace misalignments ([Galkin et al., ICML 2024](https://arxiv.org/pdf/2310.04562)), making such explorations worthy of their own research and it is impossible to complete during rebuttal period. What we can provide, is **GIST outperformed pre-trained graph models**, as showcased in  **Table 5 in our paper**.
>
>
> On the task end, our method primarily targets graph classification as clearly stated in various places of our writing. While we appreciate the reviewer’s suggestion for other graph tasks, **we must note there are countless research focusing solely on graph classification advancement, and contribution in this regard is well-recognized.**
>
> Still, to **showcase generalizability**, we present results on **two node-level datasets and one large-scale graph regression task** (see **Table 1** in W2 of our response to reviewer `1uvt`). We hope this additional evidence will prompt the reviewer to reconsider the assessment.

---

> > ### Comment · Reviewer_nG8X · 2025-04-04
> >
> > Most concerns have been addressed, and the score has been raised accordingly.

---

### Official Review · Reviewer_L3nR · 2025-03-14

**Overall Recommendation:** 2

**Summary:**

This paper is aimed to effectively encode graph structure into formation within the attention mechanism. Authors propose a new structural encoding based on Graph Invariant Structural Trait (GIST) to capture substructures within a graph by estimating pairwise node intersections. Both theoretical analysis and empirical results indicate the effectiveness of the proposed GIST.

**Claims And Evidence:**

Yes, the experiments present convincing results.

**Essential References Not Discussed:**

N/A

**Experimental Designs Or Analyses:**

Most of the experimental designs are reasonable.

**Methods And Evaluation Criteria:**

Yes, the proposed method in this paper can make sense.

**Other Comments Or Suggestions:**

There are several typos, such as:

Line 204, “requires first compute the cardinality of xxx” -> “requires to first compute the cardinality of xxx”.

The character of “graph Transformer” is not consistent in the whole paper.

**Other Strengths And Weaknesses:**

Strengths:
(1) This paper introduces a new method referred to GIST that encodes graph structure using pairwise k-hop substructure vectors which can be efficiently calculated by estimating the interaction cardinality between the k-hop neighborhoods of node pairs.

(2) The movitation is clear and highlighted.

(3) Empirical results on standard graph classification benchmarks showcase consistent performance improvements, demonstrating the effectiveness of the proposed GIST.

Weaknesses:
(1) The novelty of this paper is somewhat limited, since the general idea to add structural bias to the attention mechanism is not very novel.

(2) The ablation study is insufficient due to that only the ZINC dataset is considered.

(3) Lack of efficiency experiments and parameter sensitivity analysis.

**Questions For Authors:**

(1)Can the authors provide more ablation study, such as on different datasets.

(2)Can the authors provide some efficiency experiments and the parameter sensitivity analysis?

**Relation To Broader Scientific Literature:**

This paper focuses on adding structure-aware bias to the attention mechanism in graph Transformers. It may be helpful to design effective graph Transformers.

**Theoretical Claims:**

N/A

---

> ### Author Rebuttal · Authors · 2025-04-01
>
> ### **[W1 - The novelty of this paper is somewhat limited, since the general idea to add structural bias to the attention mechanism is not very novel.]: Our novelty lies in the GIST features, which offer a more effective way to encode structural information.**
> We agree with the reviewer that adding structural bias to attention is not a new problem in graph transformers. However, it remains a **central and unresolved challenge** to determine which structural bias effectively captures complex graph structures, hence improving graph transformers. Prior works have proposed various biases, such as shortest path in Graphormer or RRWP in GRIT, yet **none have successfully captured both** substructures and the higher-order interactions between them, as we pointed out in our Motivation section. These two aspects are crucial for learning effective graph representations, as highlighted in [FragNet](https://arxiv.org/pdf/2406.08210). To the best of our knowledge, our work is **the first to address both challenges**, providing novel structural representations for effective graph representation learning in graph transformers, evidenced with our outstanding performance result.
>
> ### **[W2, W3 - Insufficient experiments on ablation study and efficiency]: Sure, here are 3 more ablation studies, training efficiency analysis, and GIST computation overhead report across 3 different datasets.**
> We thank the reviewer for raising this question. To address it, we provide the results across three datasets:
> 1. We present ablation studies on different **$k$-hops** (**Table 1**), different numbers of **MinHash functions** (**Table 2**), and different values of **HyperLogLog’s $p$** (**Table 3**). Overall, GIST demonstrates **robustness** across various hyperparameter settings. We would like to note that **higher values of HyperLogLog’s $p$** and **a greater number of MinHash functions** reduce the **error in estimating $k$-hop intersection cardinality**, leading to **improved performance** of GIST. This trend is consistently reflected in the tables.
> 2. We present an **efficiency experiment on training time** (**Table 4**), which shows that **GIST's training time is comparable to or even lower than other graph transformers**. Notably, our method **does not require extensive pretraining** like other pretrained graph models, yet it **outperforms most of them** on MoleculeNet benchmarks, as demonstrated in **Table 5 of our paper**.
> 3. An analysis of the one-time computation overhead of GIST features (**Table 4**). We would like to emphasize that **GIST computation incurs only a one-time cost at the beginning**, and this overhead is minimal. As highlighted in **Section 3.3**, we use an **efficient estimation algorithm** of GIST features using MinHash and HyperLogLog, allowing us to approximate the $k$-hop substructure intersection with a **constant number of operations**. Theoretically and empirically (Table 4), this design ensures that our method remains computationally efficient while preserving the effectiveness of structural representation.
>
> **Table 1**: Ablation study on different values of $k$-hops
> | $k$ | 1 | 2 | 3 | 4 | 5 |
> |-|:-:|:-:|:-:|:-:|:-:|
> | ZINC | 0.100 | 0.058 | 0.054 | 0.065 | 0.063 |
> | Peptides-struct | 0.2832 | 0.2471 | 0.2444 | 0.2478 | 0.2518 |
> | Peptides-func | 0.6446 | 0.6420 | 0.6790 | 0.6754 | 0.6857 |
>
> **Table 2**: Ablation study on different numbers of *MinHash functions*
> | # MinHash functions | 32 | 64 | 128 | 256 |
> |-|:-:|:-:|:-:|:-:|
> | ZINC | 0.071 | 0.069 | 0.069 | 0.058 |
> | Peptides-struct | 0.2511 | 0.2538 | 0.2447 | 0.2444 |
> | Peptides-func | 0.6502 | 0.6418 | 0.6519 | 0.6857 |
>
> **Table 3**: Ablation study on HyperLogLog data structure with different values of *p*
> | *p* | 4 | 6 | 8 | 10 |
> |-|:-:|:-:|:-:|:-:|
> | ZINC | 0.065 | 0.065 | 0.058 | 0.062 |
> | Peptides-struct | 0.2566 | 0.2545 | 0.2444 | 0.2466 |
> | Peptides-func | 0.6170 | 0.6124 | 0.6857 | 0.6771 |
>
>
> **Table 4**: One-time pre-computation and Training time of GIST (hour:min)
> | Datasets | ZINC | ZINC-full | Peptides-struct | Peptides-func |
> |-|:-:|:-:|:-:|:-:|
> | GIST precomputation | 00:03 | 01:08 | 00:12 | 00:12 |
> | GIST Training Time | 11:09 | 55:21 | 05:40 | 05:30 |
> | GRIT Training + Precomputation Time | 16:30 | 104:57 | 07:15 | 06:42 |
> | GraphGPS Training + Precomputation Time | 13:30 | - | - | - |
> | SAN Training + Precomputation Time | 32:15 | - | - | - |
>
> **Table 5**: Performance of GIST + RRWP
> | Datasets | ZINC | Peptides-struct | Peptides-func |
> |-|:-:|:-:|:-:|
> | GIST + RRWP | 0.088 | 0.2490 | 0.6453 |
> | GIST | 0.055 | 0.2442 | 0.6783 |
> | RRWP | 0.059 | 0.2460 | 0.6988 |
>
>
> ### **[W4 - Typo]**
> We thank the reviewer for carefully reading our paper. We will correct the typo in **L204** and ensure consistent usage of "graph transformers" throughout the paper.
>
> We hope the additional results and discussion can help the reviewer better understand the mechanism of our proposed method, and maybe warrent us a higher rating.

---

> > ### Comment · Reviewer_L3nR · 2025-04-06
> >
> > Thank you for the detailed responses. Some of my concerns have been addressed, but my concerns about the innovation have increased. I think the innovation is quite limited, since it seems that the substructures and interactions between nodes can be carefully captured in an existing method using the Adaptive Graph Transformer (AGT) [1]. Therefore, the core innovation may be quite limited.
> >
> > Based on the above consideration, I think this paper requires further improvement.
> >
> > [1] Ma X, Chen Q, Wu Y, et al. Rethinking structural encodings: Adaptive graph transformer for node classification task[C]//Proceedings of the ACM web conference 2023. 2023: 533-544.

---

> > > ### Author Response · Authors · 2025-04-07
> > >
> > > As much as we appreciate the reviewer being open to providing an updated review — for better or worse — during the rebuttal period.
> > >
> > > ## **We must respectfully, but also firmly, note the reviewer’s assessment presents a gross misunderstanding of both works and indicates an extreme lack of familiarity with the field — supposing this review is faithfully given, which we honestly doubt.**
> > >
> > > AGT is a 2023 work with 17 cites and gated behind a paywall. After paying, we realize there are several straightforward indicators showing:
> > >
> > > * AGT targets a different task than ours.
> > > * AGT promotes a core idea we explicitly argue against.
> > > * Existing literature clearly shows AGT does not *solve* structural awareness — a widely acknowledged open problem in Graph Transformers — where our work contributes.
> > >
> > > For a good faith discussion, we present strong evidence in all 3 regards and invite the reviewer to reevaluate for proper ICML review quality.
> > >
> > > ---
> > >
> > > ### **`1. GIST and AGT focus on different tasks (Graph vs Node Classification). This is evident from just reading GIST’s abstract and AGT’s title.`**
> > >
> > > **GIST focuses on GRAPH CLASSIFICATION**, as made clear by the *first sentence* of our Abstract and Introduction:
> > >
> > > > Graph classification is a core ML task...
> > > > Graph classification is a fundamental problem...
> > >
> > > with countless explicit statements like the following:
> > >
> > > > ... GIST effectively captures structural information critical for graph classification.
> > > > RQ 1: How well does GIST facilitate the learning and differentiation of substructures in graph classification tasks?
> > >
> > > In contrast, **AGT focuses on NODE CLASSIFICATION**, as its title makes clear:  *"Adaptive Graph Transformer for **Node Classification Task**"*
> > >
> > > Notably, AGT itself highlights this distinction:
> > >
> > > > *"... recent GTs mainly focus on graph-level tasks like... (i) What kind of information is needed for the node-level tasks?... (iii) How to design powerful GTs for the node-level tasks?"*
> > >
> > > **This exact comment from the AGT authors alone dismisses the notion that AGT invalidates graph-level studies like ours.**
> > >
> > > ---
> > >
> > > ### **`2. AGT aggregates SIMILAR substructures, while we focus on DIVERSE ones — a conceptual contrast repeatedly emphasized and recognized by all other reviewers.`**
> > >
> > > Even if one entertains a technical comparison (setting task aside), AGT and GIST adopt fundamentally different philosophies.
> > >
> > >
> > > AGT argues it is best to learn from **SIMILAR substructures**:
> > >
> > >
> > > > *"We propose ... to adaptively enhance the message exchange between nodes with high structural similarity."*
> > > > *"For node pairs with low structural similarity, the connection would be weakened..."*
> > >
> > > In contrast, GIST highlights the importance of learning from **DIVERSE substructures**, with paragraph like:
> > >
> > > > **Challenge 2. Aggregating Diverse Substructures Information**
> > >
> > > and explicit statements like:
> > >
> > > >  ... it is equally important for structural encodings to enable the aggregation of information across diverse substructures, rather than restricting it to similar or localized patterns.
> > > > ... highlights how different substructure compositions lead to distinct intersection patterns, enabling...
> > >
> > > **This difference is major, clear, and recognized by all other reviewers. In our opinion, it is `impossible to miss` for any reasonable reader who gives even minimal attention to both works.**
> > >
> > > ---
> > >
> > > ### **`3. Structural awareness (SA) in Graph Transformers (GT) is far from solved — AGT contributes, but does not disqualify future work in this direction.`**
> > >
> > >
> > > 1. Numerous publications since AGT continue to explore structural awareness/encoding (SA/SE) for GTs. Such as GRASS, MoSE (ICLR25); S2GNN, HDSE, N2C-Attn (NeurIPS24); Subgraphormer, FragNet, CoBFormer (ICML24); GRIT (ICML 23).
> > >
> > > 2. Several works explicitly call out the SE challenge as unsolved. E.g., *"Graph Positional and Structural Encoder,"* a 24 June paper from Rampášek's lab — who first-authored the well-recognized GraphGPS — states:
> > > > *"...designing ... structural encodings that work optimally for ... is a challenging and unsolved problem..."*
> > >
> > > This confirms the ongoing relevance of research like ours. GIST takes a **large and positive step** by being the first to introduce **intersection feature-based SE** for GTs.
> > >
> > >
> > >
> > > ---
> > >
> > > Thus, **the reviewer’s assessment**
> > >
> > > > *"it seems that the substructures and interactions between nodes can be carefully captured in an existing method using AGT. Therefore, the core innovation may be quite limited."*
> > >
> > >
> > > **would serve as a reason to reject most, if not all of the listed works that contribute to SA in GTs.**
> > >
> > > ## **In the most respectful way possible, this argument fails even the most basic level of sanity check and should not appear in the review process of ICML.**
> > >
> > > We respectfully invite the reviewer to **revisit** or provide **more detailed evidence** articulating how exactly AGT limits the innovation of our work — and by extension, a significant body of research in SA for GTs post AGT's appearance.

---

### Decision · Program_Chairs · 2025-05-01

**Decision:**

Reject

**Comment:**

This submission introduces Graph Invariant Structural Trait (GIST), a novel structural encoding method designed to improve Graph Transformers by encoding structural information through pairwise k-hop neighborhood intersection feature vectors. The authors demonstrate empirical improvements on standard graph classification benchmarks. This is a borderline paper receiving mixed reviews. I personally read the paper and have the following concerns: 1) The innovation in methodology is not ground-breaking. There are numerous structural features that can be integrated into GTs/GNNs to improve their performance, and existing works have been around this for years (like RWSE, Laplacian PE, triangles, paths, k-hop motifs, SPDs, Resistance distances, graph bi-connectivity, etc.). Why does neighborhood intersection particularly superior than others? Does every manual structure feature worth publishing a top-conference paper? 2) Most existing works have at least theoretical results proving the expressivity improvement of the manual features. This paper lacks such results, with only a straightforward result proving the invariance of the proposed feature. Based on the above, room is reserved for more ground-breaking papers for the graph community.